# Diversity of the gut microbiome in three grasshopper species using 16S rRNA and determination of cellulose digestibility

Jian-Mei Wang, Jing Bai, Fang-Yuan Zheng, Yao Ling, Xiang Li, Jing Wang, Yong-Chao Zhi and Xin-Jiang Li

The Key Laboratory of Zoological Systematics and Application, College of Life Science, Institute of Life Science and Green Development, Hebei University, Baoding, Hebei, China

## ABSTRACT

**Background:** Grasshoppers are typical phytophagous pests, and they have large appetites with high utilization of plants fibers, the digestion of which may depend on the microorganisms in their intestines. Grasshoppers have the potential to be utilized in bioreactors, which could improve straw utilization efficiency in the future. In this study, we describe the gut microbiome in three species of grasshoppers, *Oedaleus decorus asiaticus*, *Aiolopus tamulus* and *Shirakiacris shirakii*, by constructing a 16S rDNA gene library and analyzed the digestibility of cellulose and hemicellulose in the grasshoppers by using moss black phenol colorimetry and anthrone colorimetry.
**Results:** There were 509,436 bacterial OTUs (Operational Taxonomic Units) detected in the guts of all the grasshoppers sampled. Among them, Proteobacteria and Firmicutes were the most common, *Aiolopus tamulus* had the highest bacterial diversity, and *Shirakiacris shirakii* had the highest bacterial species richness. The intestinal microflora structure varied between the different species of grasshopper, with *Aiolopus tamulus* and *Shirakiacris shirakii* being the most similar. Meanwhile, the time at which grasshopper specimens were collected also led to changes in the intestinal microflora structure in the same species of grasshoppers. *Klebsiella* may form the core elements of the microflora in the grasshopper intestinal tract. The digestibility of cellulose/hemicellulose among the three species grasshoppers varied (38.01/24.99%, 43.95/17.21% and 44.12/47.62%). LEfSe analysis and Spearman correlation coefficients showed that the hemicellulosic digestibility of *Shirakiacris shirakii* was significantly higher than that of the other two species of grasshopper, which may be related to the presence of *Pseudomonas, Stenotrophomonas*, *Glutamicibacter*, *Corynebacterium*, and *Brachybacterium* in *Shirakiacris shirakii* intestinal tract.
**Conclusion:** The intestinal microbial communities of the three grasshoppers species are similar on phylum level, but the dominant genera of different species grasshoppers are different. The cellulose digestibility of the three species of grasshoppers is relatively high, which may be correlated with the presence of some gut microbiome. Increasing the understanding of the structure and function of the grasshopper intestinal microflora will facilitate further research and the utilization of intestinal microorganisms in the future.

Corresponding author
Xin-Jiang Li, lixinjiang@hbu.cn

## INTRODUCTION

Grasshoppers (Orthoptera: Acridoidea) are the main pests in agriculture, cattle grazing and forestry. Grasshoppers require a large quantity of gramineous plants to obtain the nutrients and water necessary for their survival, especially in their adult stage. The food selectivity of grasshoppers is affected by many factors. As far as plants themselves are concerned, the factors that affect grasshoppers' food selectivity include cellulose, water, carbohydrate and protein contents (*Ibanez et al., 2013*). Wheat seedlings, which have a moisture content of 89.819–93.326%, are rich in protein, vitamins, minerals and other nutrients (*Min et al., 2017*) and are easy to cultivate, making them good fodder for grasshoppers bred in laboratories.

Cellulose and hemicellulose are the main components of many biomass (*Mueller-Hagedorn & Bockhorn, 2007*). Due to the limitation of the lignin-hemicellulose, most of the biomass are difficult to be decomposed and utilized (*Thompson et al., 2003*). Many factors, like lignin content, crystallinity of cellulose, and particle size, limit the digestibility of the hemicellulose and cellulose present in the lignocellulosic biomass (*Hendriks & Zeeman, 2009*). At present, cellulose and hemicellulose are increasingly widely used (*Xiong, Zuo & Zhu, 2005*), and their efficient utilization is of great practical significance to reduce the burning of straw and promote the sustainable development of agriculture and animal husbandry.

The gut microbiome is a general term for all the microorganisms inhabiting the digestive tract of animals (*Rangberg et al., 2012*) and contains the most concentrated set of interactions among all symbiotic microorganisms in animals (*Guo et al., 2015*). In the process of evolution, insects and intestinal microorganisms interact, cooperate and coevolve. Insects secrete digestive enzymes by means of symbiotic microorganisms in the body to better digest food and obtain energy needed for their own growth and development (*Mason, Jones & Felton, 2019*). It is possible to contrive a species-wide metabolic interaction network of the termite gut-microbiome in order to have a system-level understanding of metabolic communication. *Kundu et al. (2019)* have elucidated 15 crucial hemicellulolytic microbes and their corresponding enzyme machinery (*Kundu et al., 2019*). At present, no insect has been found to be able to completely digest lignocellulose food via cellulase and hemicellulase secreted by itself (*Sun & Chen, 2010*).

Compared with termites and cockroaches, grasshoppers have a very sparse microbiome (*Dillon & Dillon, 2004*). But these microorganisms play an important role in the grasshopper digestive tract. Studies have shown that changing the structure of the intestinal microbial community can affect the survival rate of grasshoppers (*Tan et al., 2020*); *Dillon, Vennard & Charnley, 2002* have discovered that locust gut bacteria were responsible for the production of components of the locust cohesion pheromone.
At present, research on the intestinal microbial community of insects mainly focuses on certain economic insects, including silkworm, *Ceroplastes japonica*, and others, to improve

**Table 1** Information on the studied samples.

| Species | Sample code | No. of specimens | Locality | Collection date |
|---|---|---|---|---|
| *Aiolopus tamulus* | At1 | 10 | Baoding, China | 15 July 2018 |
| | At2 | 10 | Baoding, China | 15 July 2018 |
| | At3 | 11 | Baoding, China | 1 October 2018 |
| *Oedaleus decorus asiaticus* | Od1 | 10 | Baoding, China | 1 October 2018 |
| | Od2 | 10 | Baoding, China | 1 October 2018 |
| | Od3 | 12 | Baoding, China | 15 July 2018 |
| *Shirakiacris shirakii* | Ss1 | 10 | Baoding, China | 1 October 2018 |
| | Ss2 | 10 | Baoding, China | 1 October 2018 |
| | Ss3 | 10 | Baoding, China | 1 October 2018 |

the intestinal environment to reduce silkworm diseases or to increase the wax secretion of *Ceroplastes japonica* (*Yi et al., 2001*; *Bei, Liu & Cui, 2005*). In addition, other insects, such as ants and longicorn beetles, have been studied for their role in decomposing lignocellulose (*Zhang et al., 2005*).

There are few studies on the composition of the grasshopper intestinal microflora structure, community diversity and functional bacteria. In addition, current research is based on traditional culture methods or traditional molecular biology techniques, and grasshopper intestinal microorganisms have not yet been thoroughly investigated. In this study, the intestinal bacterial community structures of three species grasshoppers were studied by constructing a 16S rDNA library technology, and the abundance and phylogenesis of these bacteria were analyzed to obtain better information on grasshopper intestinal microbial diversity, providing a theoretical basis for clarifying the mechanism of cellulose degradation in grasshopper, and further study the relationship between intestinal microorganisms and pest control. At the same time, the digestibility of cellulose and hemicellulose in the grasshoppers were determined by using moss black phenol colorimetry and anthrone colorimetry, providing basic data for the development of a cellulose and hemicellulose digestion bioreactor, as well as a feasible method for determining insects' cellulose and hemicellulose digestibility.

# MATERIALS AND METHODS

## Specimen collection

Adults of *Oedaleus decorus asiaticus* Bey-bienko, 1941, *Aiolopus tamulus* Fabricius, 1789 and *Shirakiacris shirakii* Bolívar, 1914, were collected from Baoding City, Hebei Province, China in July–November 2018 (Table 1).

## Intestinal microbial diversity of grasshoppers

Total DNA of the intestinal contents of the three species grasshoppers was extracted, with each species having three groups of samples, totaling 9 sample groups. The sample numbers are shown in Table 1. Total DNA of the nine sample groups was used as templates, and PCR was carried out with universal primers targeting the 16S rDNA V3+V4

region of prokaryotes. After the PCR products passed quality tests, they were detected by an Illumina HiSeq 2500 sequencer (at Biomarker Technologies Corporation), and the data were processed and analyzed by Uparse and QIIME software (*Caporaso et al., 2010*).

### Sample treatment

The collected and classified living grasshoppers were placed in cages without access to food for 2 days to remove their intestinal contents. The grasshoppers to be tested were washed repeatedly with sterile water, placed in a 75% alcohol solution for 2 min, washed with sterile water, irradiated with ultraviolet light for 3–5 min, and dissected grasshoppers under sterile conditions. The entire intestinal tract was removed, and the midgut and hindgut parts were separated; placed in labeled, sterilized 1.5 mL centrifuge tubes; and kept at −80 °C for later use.

### Extraction of total DNA from the intestinal contents

Total DNA of the intestinal contents of grasshoppers was extracted using the PowerSoil DNA Isolation Kit according to the manufacturer's protocol, and the quality and quantity of DNA were evaluated by the 260 nm/280 nm and 260 nm/230 nm ratios, respectively. DNA was then stored at −80 °C until further processing.

For each individual sample, the 16s rDNA V3 + V4 region was amplified using the 338 F (5′-ACTCTACGGAGAGCA-3′) and 806 R (5′-GGACTACHVGGGTWTCTAT-3′) primers (*Mori et al., 2014*). PCR was performed in a total reaction volume of 20 μL: $H_2O$, 13.25 μL; 10×PCR ExTaq Buffer, 2.0 μL; DNA template (100 ng/mL), 0.5 μL; primer1 (10 mmol/L), 1.0 μL; primer2 (10 mmol/L), 1.0 μL; dNTP, 2.0 μL; and ExTaq (5 U/mL), 0.25 μL. After an initial denaturation at 95 °C for 5 min, amplification was performed with 30 cycles of incubations for 30 s at 95 °C, 20 s at 58 °C and 6 s at 72 °C, followed by a final extension at 72 °C for 7 min. The amplified products were then purified and recovered using 1.0% agarose gel electrophoresis. Finally, all the PCR products were quantified by Quant-iT$^{TM}$ dsDNA HS Reagent and pooled together. High-throughput sequencing analysis of bacterial rRNA genes was performed on the purified, pooled samples using the Illumina HiSeq 2500 platform (2 × 250 pairedends) at Biomarker Technologies Corporation, Beijing, China. Finally, library construction and sequencing were performed by Beijing Biomarker Technologies Co. Ltd.

### Bioinformatics analysis

Bioinformatics analysis in this study was completed on the Biomarker Cloud Platform (Biomarker Biotechnology Co., Beijing, China). The original data obtained by sequencing were spliced by FLASH software. Then, raw tags were filtered and clustered. Sequences were removed from inclusion according to the following criteria: the average mass of bases was less than 20; the reads were low quality; the sequences contained primer mismatches; the sequences were less than 350 bp in length; and the sequences could not be spliced. UCHIME, a tool included in mothur (http://drive5.com/uchime), was used to remove chimeras and generate valid data. OTUs were taxonomically annotated based on the Silva (bacteria) and UNITE (fungi) taxonomic databases. The denoised sequences were clustered using USEARCH (version 10.0), and tags with similarity ≥97% were regarded as

OTUs. Taxonomy was assigned to all OTUs by searching against the Silvadatabases (http://www.arb-silva.de.) using uclust within QIIME (*Edgar, 2010*).

## Digestibility of wheat seedlings in grasshoppers

### *Collection and treatment of samples*

Grasshoppers collected in the field were separately packed in insect rearing cages, and each cage contained 10 individuals that were fed wheat seedlings (The wheat variety was *Triticum aestivum* Linnaeus, 1753). After consecutively feeding for 3 days (no dung was collected during the period, and the wheat seedlings provided sufficient nutrition), grasshoppers were fasted for 2 days. A layer of white plastic foam was spread on the bottom of the cages to facilitate the collection of excrement (*Wang, Liu & Chen, 2008*). During the experiment, the fresh weight of wheat seedlings fed each time was recorded, and the feces and residual wheat seedlings were dried to a constant weight at 70 °C and recorded (using an electrothermal constant temperature blast drying oven, DGG-9030A; Shanghai Flyover Experimental Instrument Co., Ltd.). The dry-fresh ratio of wheat seedlings was determined to calculate the dry weight of the wheat seedlings before the experiment (*Wang, 1997*). The collected feces were dried to a constant weight, pulverized, and filtered with a 40 mesh sieve.

The wheat seedlings were rapidly dehydrated by steam de-enzyming (*Sun, 2014*), dried at 70 °C until a constant weight, crushed, and filtered with a 40 mesh sieve for later use.

### *Determination of cellulose and hemicellulose content*

Samples were prepared by weighing out 0.800 g of each sample, to which 8 mL 72% $H_2SO_4$ was added, followed by shaking. Samples were placed in a water bath at 30 °C for 1 h, followed by the addition of eight mL 4% $H_2SO_4$, and were then returned to the water bath for 45 min. Finally, 224 mL of distilled water was added, and the samples shaken well before being placed into conical flasks in an electric heating pressure steam sterilization pot (LS-30 type of Shanghai Bosun Industrial Co., Ltd.). Samples were then heated to a temperature of 121 °C for 1 h and filtered to obtain sample solutions.

One milliliter of this sample solution was diluted appropriately, and one mL of the diluted sample solution was added to one mL of anthrone reagent and three mL of 80% sulfuric acid, mixed well, and boiled at 100 °C for 5 min. After cooling to room temperature, absorbance at 620 nm was measured, with the sugar concentration calculated according to the glucose standard regression equation and then multiplied by 0.9 (*Zhang et al., 2010*).

One milliliter of the sample solution was diluted appropriately, and one mL of the diluted sample solution was add to two mL of A reagent and 0.134 mL of B reagent and boiled at 100 °C for 20 min after fully mixing. Absorbance at 660 nm was measured after cooling to room temperature, with the sugar concentration calculated according to the xylose standard regression equation and then multiplied by 0.88 (*Zhang et al., 2010*).

### *Calculation of the decomposition rates of cellulose and hemicellulose*

The decomposition rates of cellulose and hemicellulose were calculated after the cellulose and hemicellulose contents of the adult grasshopper feces were determined by the above

Table 2 Sequence and proportion results of each sample and bacterial identification results.

| Sample | Clean tags | Effective tags | Proportion (%) | Identification result |
|---|---|---|---|---|
| At1 | 53704 | 53325 | 99.29 | 4 Phyla, 7 classes, 11 orders, 18 families, 26 genera |
| At2 | 51643 | 51479 | 99.68 | 5 Phyla, 9 classes, 13 orders, 24 families, 31 genera |
| At3 | 61063 | 61018 | 99.93 | 5 Phyla, 10 classes, 17orders, 28 families, 28 genera |
| Od1 | 61047 | 61024 | 99.96 | 6 Phyla, 9 classes, 12 orders, 18 families, 21 genera |
| Od2 | 72346 | 72296 | 99.93 | 6 Phyla, 9 classes, 16 orders, 23 families, 27 genera |
| Od3 | 53117 | 52034 | 97.96 | 5 Phyla, 7 classes, 11 orders, 17 families, 22 genera |
| Ss1 | 52796 | 52631 | 99.68 | 5 Phyla, 9 classes, 16 orders, 28 families, 32 genera |
| Ss2 | 53296 | 53144 | 99.71 | 5 Phyla, 8 classes, 15 orders, 25 families, 31 genera |
| Ss3 | 53097 | 52485 | 98.85 | 5 Phyla, 8 classes, 13 orders, 24 families, 30 genera |
| Total | 512109 | 509436 | 99.48 | 7 Phyla, 12 classes, 20 orders, 42 families, 54 genera |

methods. Statistical analysis of digestibility data was done in SPSS 21.0 software using *T*-test.

$$\text{cellulose (hemicellulose)} = \frac{c \times 240 \times 10^{-3} \times 0.9(0.88)}{m} \times \text{dilution mutiple} \times 100\%$$

$$\text{cellulose digestibiliy} = \frac{a - b}{a} \times 100\%$$

Note: $c$ is the sugar concentration (g/L) calculated according to the standard curve, $m$ is the weighed sample mass (g). $a$ is amount of cellulose fed on wheat seedlings (g), $b$ is fecal cellulose content (g).

### Correlation between digestibility and microorganism abundance

The LefSe analysis and Spearman analysis were performed using R and the Psych, Pheatmap and reshape2 package (*Kostic et al., 2015*) on the Biomarker Cloud Platform. The correlation between cellulose digestibility and intestinal microbial diversity of grasshoppers was established.

## RESULTS

### Intestinal microbes in grasshoppers

#### *Evaluation of sequencing quality*

A total of 702,445 paired-end reads were obtained by sequencing the nine pooled samples. and 512,109 clean tags were generated after splicing and filtering the paired-end reads. A minimum of 51,643 clean tags were generated for each sample, with an average of 56,901 clean tags. The proportion of effective sequences was 99.48%. The sequencing accuracy of the samples was high and met the standard requirements. Effective tags were the number of effective sequences after filtering chimeras from the clean tags. The number of sequences and the proportion for each sample are shown in Table 2 below.

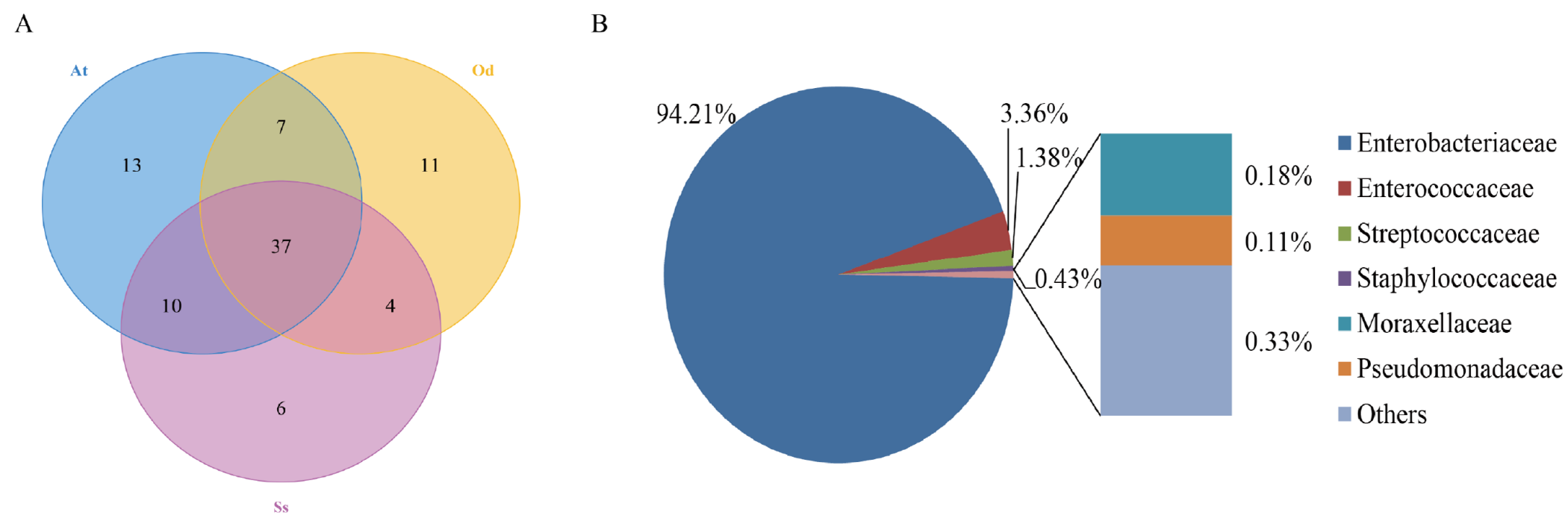

**Figure 1 The results of OTU-Venn analysis.** (A) OTU Venn diagram among different species. (B) Composition of common OTUs at the family level.

### OTU-venn analysis

To identify the number of common and unique OTUs among samples, a Venn diagram was used, which intuitively reflects the coincidence of OTUs among samples. As shown in Fig. 1A, there were 37 species of bacteria in the intestinal tract common to the three species of grasshoppers. There were six species specific to *Shirakiacris shirakii*, 11 species specific to *Oedaleus decorus asiaticus* and 13 species specific to *Aiolopus tamulus*. Further analysis of the identifies of the bacteria common to the three grasshopper species indicated that they were mainly composed of two families of Enterobacteriaceae and Enterococcaceae, as shown in Fig. 1B, with a relative abundance of 97.57%, indicating that these two families may form the core microflora in the grasshopper intestinal tract.

### $\alpha$-diversity analysis

As shown in Fig. 2A, the rarefaction curves of 9 samples tended to be flat over an increasing number of sequences. The Shannon, Simpson, Chao1 and ACE indices, as well as others, were used to express the $\alpha$-diversity of the microorganisms in the samples. As shown in Table 3, the coverage of the nine samples was relatively high, reaching 99.97~99.99%. The above results show that the sequencing data were reasonable and that the vast majority of bacteria in the samples were detected. Different from the rarefaction curve, the species accumulation curve reflects whether the number of samples was sufficient and whether the information covered all the annotated species. As shown in Fig. 2B, as the sample number increased, the cumulative curve and the common quantity curve tended to be flat, which demonstrates that the new and common species detected in the sample were both approaching saturation, indicating that the sample size was sufficient and could be used for diversity and abundance analysis.

The $\alpha$-diversity of nine samples varied according to the individual. In the three samples of *Aiolopus tamulus*, the Shannon index of At1 and At2 was much higher than that of At3, while the Simpson index of At3 was the opposite, indicating that the species diversity in samples At1 and At2 was higher than that in At3. Among the three samples of *Oedaleus decorus asiaticus*, the Shannon index of Od3 was much higher than that of Od1 and Od2, while the Simpson index of Od3 was much lower than that of the other two

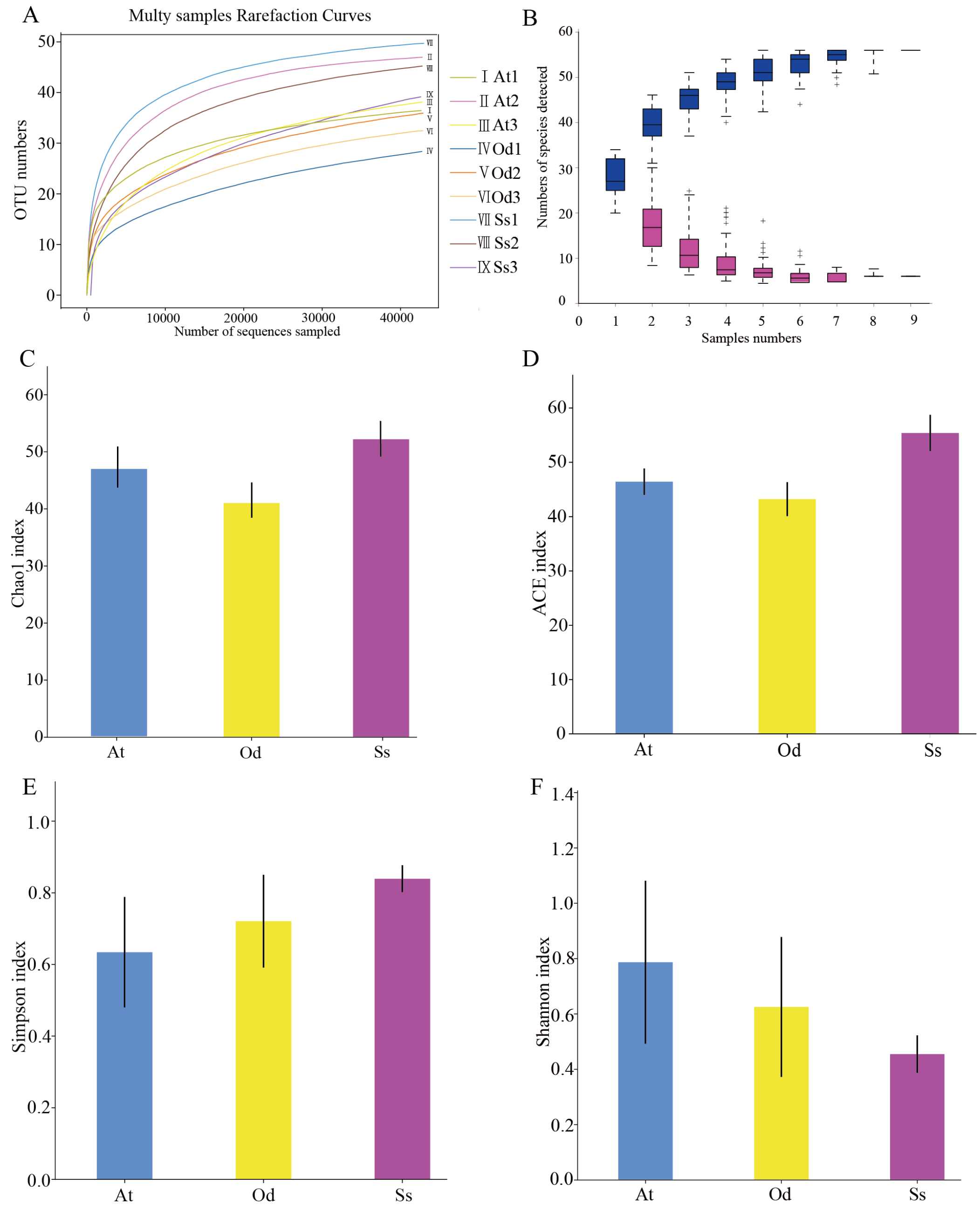

**Figure 2** **The results of α-diversity analysis.** (A) Rarefaction curves of 9 samples of grasshopper intestinal contents. (B) Species discovery curve. (A single blue box in this figure represents the total number of species detected in randomly selected samples. The cumulative curve is composed of the totality of blue boxes, which represents the rate of new species appearing under continuous sampling; a single purple box in this figure represents the number of common species detected in a given number of samples. The set of purple boxes form the common quantity curve, which represents the rate of common species detected under continuous sampling). (C) Chao1 index of the three species grasshoppers. (D) ACE index of the three species grasshoppers. (E) Simpson index of three species grasshoppers. (F) Shannon index of the three species grasshoppers.

**Table 3** Statistical results of the diversity index of the intestinal content samples of grasshoppers.

| Simple ID | OTU | ACE | Chao1 | Simpson | Shannon | Coverage |
|---|---|---|---|---|---|---|
| At1 | 37 | 41.5776 | 40.0000 | 0.3996 | 1.1721 | 0.9999 |
| At2 | 47 | 48.7316 | 48.2000 | 0.5780 | 0.9800 | 0.9999 |
| At3 | 41 | 48.9659 | 52.2500 | 0.9249 | 0.2079 | 0.9998 |
| Od1 | 31 | 41.2173 | 38.2000 | 0.9211 | 0.2320 | 0.9998 |
| Od2 | 40 | 49.3557 | 47.2000 | 0.7624 | 0.5464 | 0.9999 |
| Od3 | 33 | 39.0695 | 37.6667 | 0.4783 | 1.0977 | 0.9998 |
| Ss1 | 50 | 51.9067 | 50.8571 | 0.8528 | 0.4748 | 0.9999 |
| Ss2 | 46 | 52.1871 | 51.6000 | 0.8964 | 0.3283 | 0.9998 |
| Ss3 | 44 | 62.0907 | 54.1111 | 0.7679 | 0.5609 | 0.9997 |

samples. For the three samples of *Shirakiacris shirakii*, the Shannon index and Simpson index were not significantly different, which may be related to the difference in the collection time (Table 1).

The average value of each index of three samples from the same species was calculated and then used to compare and analyze the α-diversity among the different species. The Chao1 index (Fig. 2C) of *Shirakiacris shirakii* was significantly higher than that of *Oedaleus decorus asiaticus*, and the ACE index (Fig. 2D) was the highest in *Shirakiacris shirakii*, followed by *Aiolopus tamulus*, which demonstrates that among the three species grasshoppers, the abundance of species in the intestinal tract of *Shirakiacris shiraki* was significantly higher than that of *Oedaleus decorus asiaticus*, with *Aiolopus tamulus* in the middle. The Simpson index (Fig. 2E) of *Aiolopus tamulus* was the smallest, while the index of *Shirakiacris shiraki* was the largest. The Shannon index (Fig. 2F) followed the opposite trend to the Simpson index, which indicated that the species diversity in the intestinal tract of *Aiolopus tamulus* was the highest, followed by the *Oedaleus decorus asiaticus*, with *Shirakiacris shiraki* as the lowest.

### *β-diversity analysis*

Based on pyrosequencing data, PCoA and UPGMA clustering were carried out to determine β-diversity. As shown in Fig. S1, the smaller the distance between points in the figure, the smaller the difference in the intestinal flora structure, and vice versa. It can be seen from the figure that the difference in the intestinal microflora structure between the three samples of *Shirakiacris shiraki* and two of the samples of *Aiolopus tamulus* was relatively small, while difference in the intestinal microflora structure between one sample and the remaining two samples for both *Oedaleus decorus asiaticus* and *Aiolopus tamulus* was relatively large. The difference in the intestinal microflora structure among the three samples of *Shirakiacris shiraki* was not large. In addition, the hierarchical cluster tree (Fig. 3A) shows that the microbial communities of the three species grasshoppers are divided into three groups: (1) group I includes samples A1 and A2 and sample O3, (2) group II includes samples O1 and O2 and sample A3 and (3) group III includes all the samples of *Shirakiacris shiraki*. In addition, the distance between group II

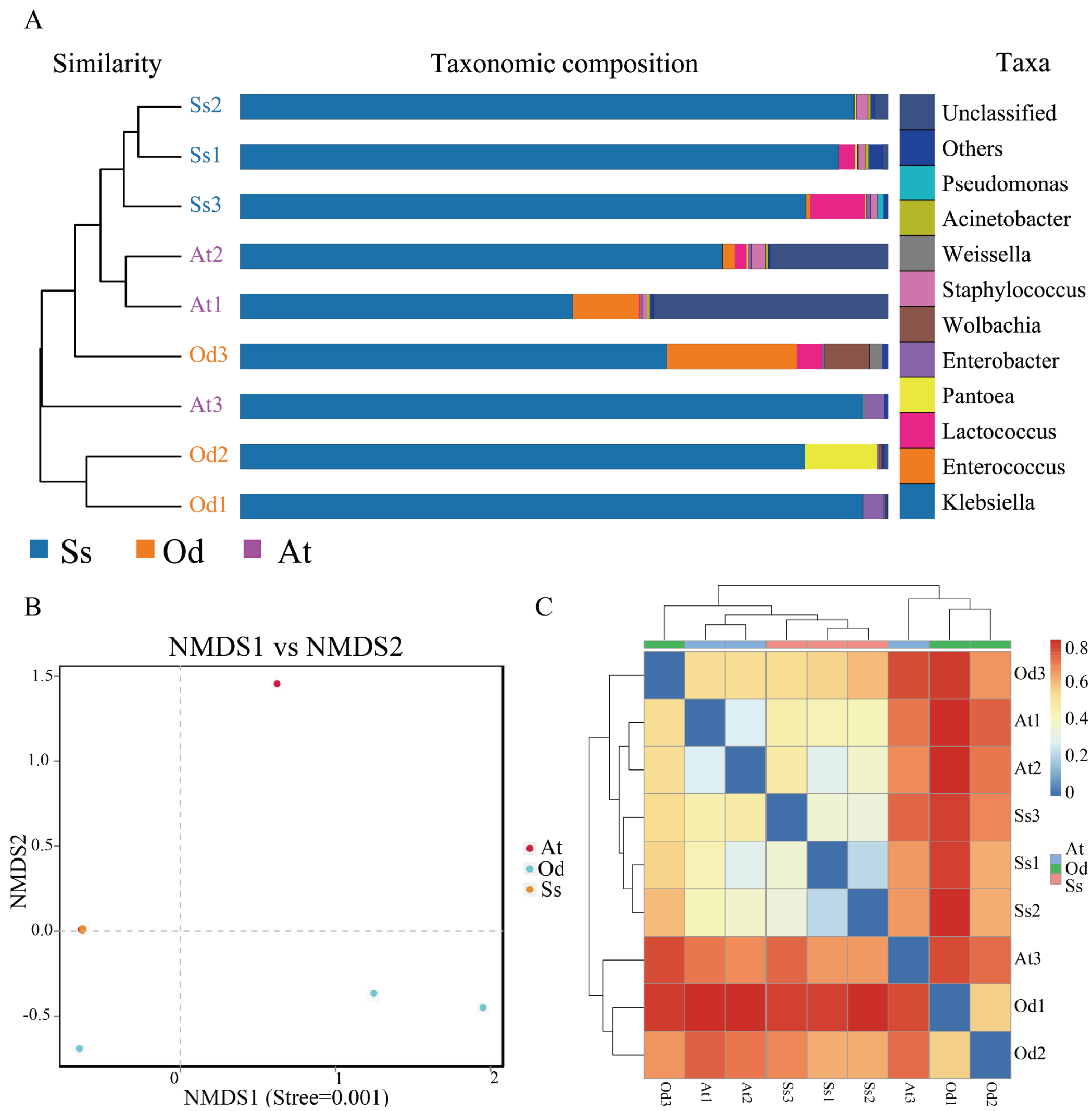

**Figure 3 The results of β-diversity analysis.** (A) UPGMA cluster analysis of the three species grasshoppers. (B) NMDS analysis based on the binary-jaccard distance. (C) Heatmap of each sample at the OTU classification level (Blue indicate similarity and red indicate distance).

and group III was closer, that is, the composition of the intestinal microflora is more similar between those two groups. Taken together, these results show that the intestinal microflora of different species of grasshoppers vary from one another. The intestinal microflora of *Aiolopus tamulus* and *Shirakiacris shiraki* are more similar. At the same time, different sampling times will also lead to the recombination of microbial communities.

Nonmetric Multidimensional Scaling (NMDS) analysis can reflect the differences between groups or within groups according to the distribution of samples. As shown in Fig. 3B, the stress value is less than 0.01, which indicates that the analysis result is extremely reliable. In the figure, it can be seen that there is a large difference in the intestinal community between one sample the remaining two samples for both *Oedaleus decorus asiaticus* and *Aiolopus tamulus*, which is related to the different collection times of

the samples, indicating that a difference in collection time leads to changes in the microbial community structure of the same species. The three samples of *Shirakiacris shiraki* along with two samples of *Aiolopus tamulus* are almost coincident, which indicates that the similarity of the intestinal microflora structure between the two groups was relatively high.

As shown in Fig. 3C, Ss1, Ss2 and Ss3 were grouped together; At1 and At2 were grouped together; and all (Ss1, Ss2, Ss3 and At2) were grouped with Od3. Od1 and Od2 were grouped with At3. The samples At1 and At2 of *Aiolopus tamulus* were relatively close to the three samples of *Shirakiacris shiraki*, which indicates that the intestinal community similarity between *Aiolopus tamulus* and *Shirakiacris shiraki* is high, that the difference of the microflora structure between them is relatively small, and that different collection times for the same species can lead to low similarity and large differences in the grasshopper intestinal microflora structure, which is consistent with the above results, indicating that a difference in collection time causes changes in the microbial community structure.

### Intestinal microflora structure of the three species grasshoppers

High-quality sequences obtained from 16S rDNA identification were compared with the database, and a total of 54 genera of seven phyla, 12 classes and 20 orders were identified. The composition of each sample is shown in Table 2. Once the average relative abundance of different grasshoppers in the same treatment at each classification level is calculated, the average relative abundance can reflect the content of various intestinal microorganisms at the overall level.

### Intestinal microflora structure at the phylum level

The nine samples At1, At2, At3, Od1, Od1, Od2, Od3, Ss1, Ss2 and Ss3 contained 85.65%, 83.51%, 93.45%, 89.51%, 91.43%, 87.32%, 86.92%, 87.33% and 87.35%, respectively, of the valid sequences that were able to be annotated at the phylum level. Seven phyla were detected in the nine samples. According to the annotation results of the samples at various classification levels (kingdom, phyla, class, order, family, genus and species), as shown in Fig. 4A, Proteobacteria accounted for the highest relative abundance in the three species of grasshoppers, *Aiolopus tamulus, Oedaleus decorus asiaticus* and *Shirakiacris shiraki*, at 94.10%, 90.72% and 93.94%, respectively. The second highest was Firmicutes, accounting for 5.72%, 8.94% and 5.31%, respectively.Actinobacteria accounted for a relatively high proportion of 0.52% in the intestinal tract of *Shirakiacris shiraki*, although less than 0.10% in the intestinal tracts of the other two species. Cyanobacteria was relatively abundant in the intestinal tract of *Oedaleus decorus asiaticus*, at 0.26%, while its abundance in the other two species was very small, accounting for 0.01%. Fusobacteria existed in the intestinal tracts of the three species grasshoppers in trace amounts, accounting for less than 0.10%. Bacteroidetes was found in trace amounts in *Aiolopus tamulus* and *Oedaleus decorus asiaticus* but was not detected in the intestinal tract of *Shirakiacris shiraki*. Tenericutes was only found in trace amount in the intestinal tract of *Oedaleus decorus asiaticus*, at 0.04%, but was not found in the intestinal tracts of the other two grasshoppers. Additionally,

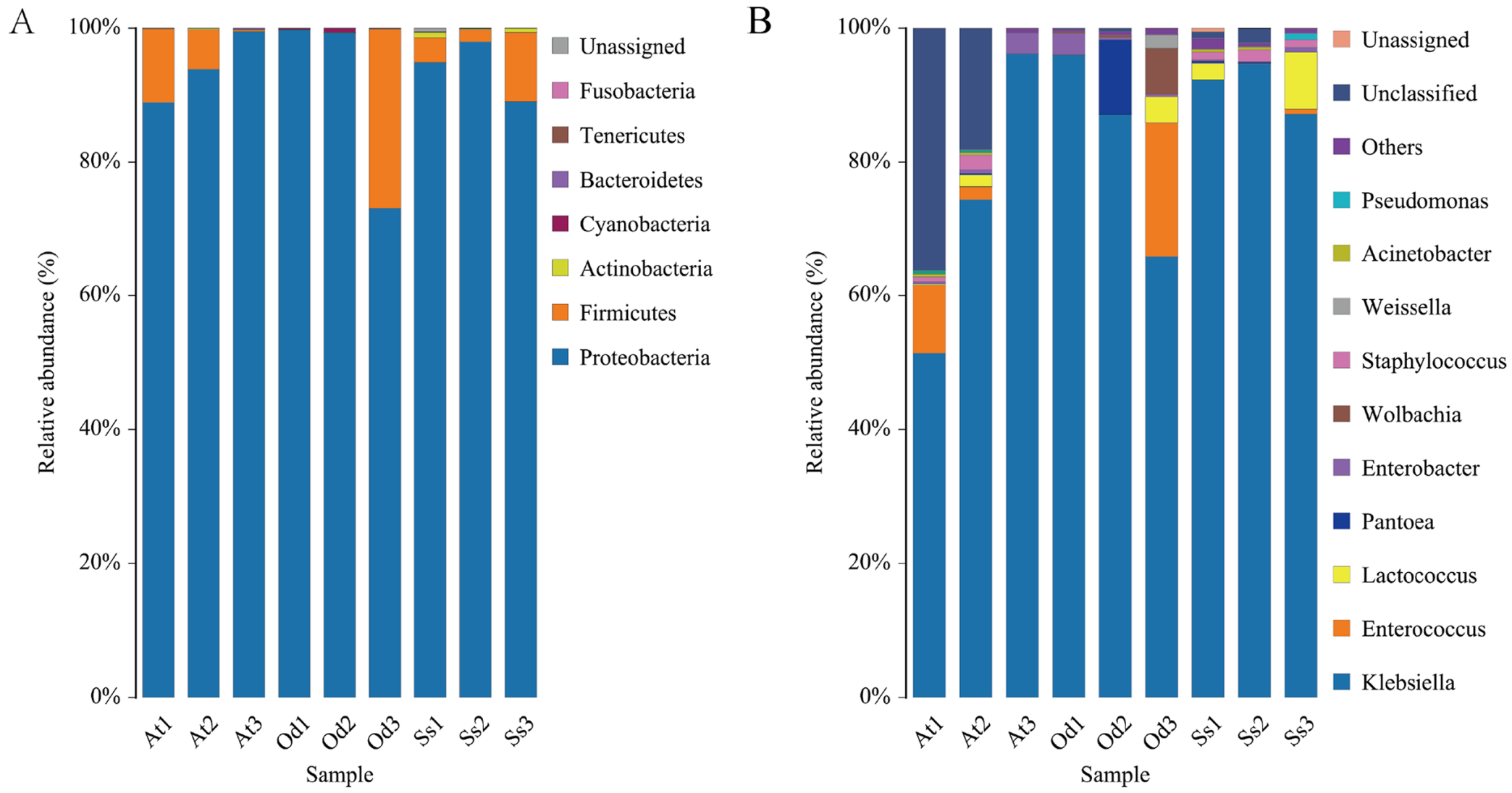

**Figure 4 Diversity of the bacterial microbiota in the three species grasshoppers guts at the phylum (A)/genus (B) level.**

0.20% unassigned microorganisms were present in the intestinal tract of *Shirakiacris shiraki* that have not previously been studied.

It is worth noting that the proportion of Firmicutes in At3 intestinal bacteria was 0.24%, which was much lower than that in At 1 (11.01%) and At2 (5.91%) treated with the same method. However, the proportion in sample Od3 (26.74%) was much higher than that in sample Od1 (0.03%) and sample Od2 (0.06%), while the proportion in the three samples Ss1 (3.72%), Ss2 (1.83%) and Ss3 (10.36%) of *Shirakiacris shiraki* was relatively constant, which could be related to their different collection times, indicating that the abundance of intestinal flora varied over different periods in the same species. Combined with α-diversity analysis, these results show that the diversity and abundance of intestinal microflora varied over different periods in the same species.

### Intestinal microflora structure at genus level

At1, At2, At3, Od1, Od2, Od3, Ss1, Ss2 and Ss3 contained 54.57%, 68.35%, 93.40%, 89.35%, 90.93%, 87.31%, 86.08%, 85.45% and 87.30%, respectively, of the valid sequences that could be annotated at the genus level. A total of 54 bacterial genera were detected, of which 24 bacterial genera were common among the three species. As seen in Fig. 4B, *Klebsiella* accounted for the highest proportion of the microbial community in the three grasshopper species. The top 10 abundant bacterial genera (by average relative abundance) for each of the three species of grasshoppers after data standardization are shown in Tables 1–3. In the three samples of *Aiolopus tamulus*, the average relative abundance of *Klebsiella*, *Enterococcus* and *Enterobacter* was greater than 1%, which identifies them as the primary bacteria in the *Aiolopus tamulus* intestinal tract. The primary bacteria of *Oedaleus decorus asiaticus* were *Klebsiella*, *Enterococcus*, *Pantoea*, *Wolbachia*, *Enterobacter* and *Lactococcus*. *Klebsiella*, *Lactococcus* and *Staphylococcus* were the primary genera of *Shirakiacris shiraki*. Five bacterial genera were detected only in the intestinal tract of

**Table 4 Digestibility of cellulose and hemicellulose in wheat seedlings in three species grasshoppers.**

| Sample | Digestibility rate of cellulose | Digestibility rate of hemicellulose |
|---|---|---|
| *Aiolopus tamulus* | 43.95 ± 2.02a | 17.21 ± 2.98b |
| *Oedaleus decorus asiaticus* | 38.01 ± 3.96a | 24.99 ± 4.80b |
| *Shirakiacris shirakii* | 44.12 ± 3.60a | 47.65 ± 3.37a |

**Note:**
The data in the table are expressed as the mean standard error, and the data in the same column with different lowercase letters show significant difference ($P < 0.05$).

*Aiolopus tamulus*, namely, *Anaerotruncus*, *Diaphorobacter*, *Morganella*, *Proteiniclasticum* and *Rikenellaceae*_RC9_gut_group. The proportion of these five bacterial genera in the intestinal tract was not more than 0.1%. Among them, *Morganella* was not detected in the At3 samples, but was detected in the At1 and At2 samples, and the remaining four genera were detected only in the At3 samples but not in the At1 and At2 samples, indicating that there were significant differences in the intestinal microflora diversity of the same species from different time periods. The genera *Sphaerotilus* and *Spiroplasma* were only detected in the intestinal tract of *Oedaleus decorus asiaticus*, and *Cronobacter* was only detected in *Shirakiacris shiraki*. Therefore, the diversity of the intestinal microorganisms varied by grasshopper species.

## Digestibility results

From Table 4, the cellulose digestibility of the three species of grasshoppers were 43.95%, 38.01% and 44.12%, and there was no significant difference ($P > 0.05$) among the three groups. However, the hemicellulose digestibility in *Shirakiacris shirakii* at 47.65% was significantly higher ($P < 0.05$) than that in *Aiolopus tamulus* (17.21%) and *Oedaleus decorus asiaticus* (24.99%). In addition, the cellulose digestibility in *Aiolopus tamulus* and *Oedaleus decorus* was significantly higher than that of hemicellulose, and there was no significant difference between the cellulose and hemicellulose digestibility in *Shirakiacris shirakii*.

## Correlation between digestibility and microorganism abundance

In view of the fact that *Shirakiacris shirakii* can be distinguished from the other two species and that the digestibility of hemicellulose is significantly higher in that organism than in the other two species ($P < 0.05$), we conducted LEfSe analysis (Fig. 5; Fig. S2) on the three species and identified *Pseudomonas*, *Stenotrophomonas*, *Glutamicibacter*, *Corynebacterium*, *Brachybacterium* and other bacteria genera as biomarkers of group difference. The relative abundance of these identified species in *Shirakiacris shirakii* is significantly higher than that in the other species, which may be related to the degradation of hemicellulose. To further screen out bacteria related to the degradation rate of cellulose and hemicellulose, we calculated the Spearman correlation coefficients (Fig. 6) for the association between the degradation rate and microflora abundance and identified a number of bacteria whose abundance had a high correlation with the degradation rates of cellulose and hemicellulose. Some of the results highly overlap with the LEfSe analysis,

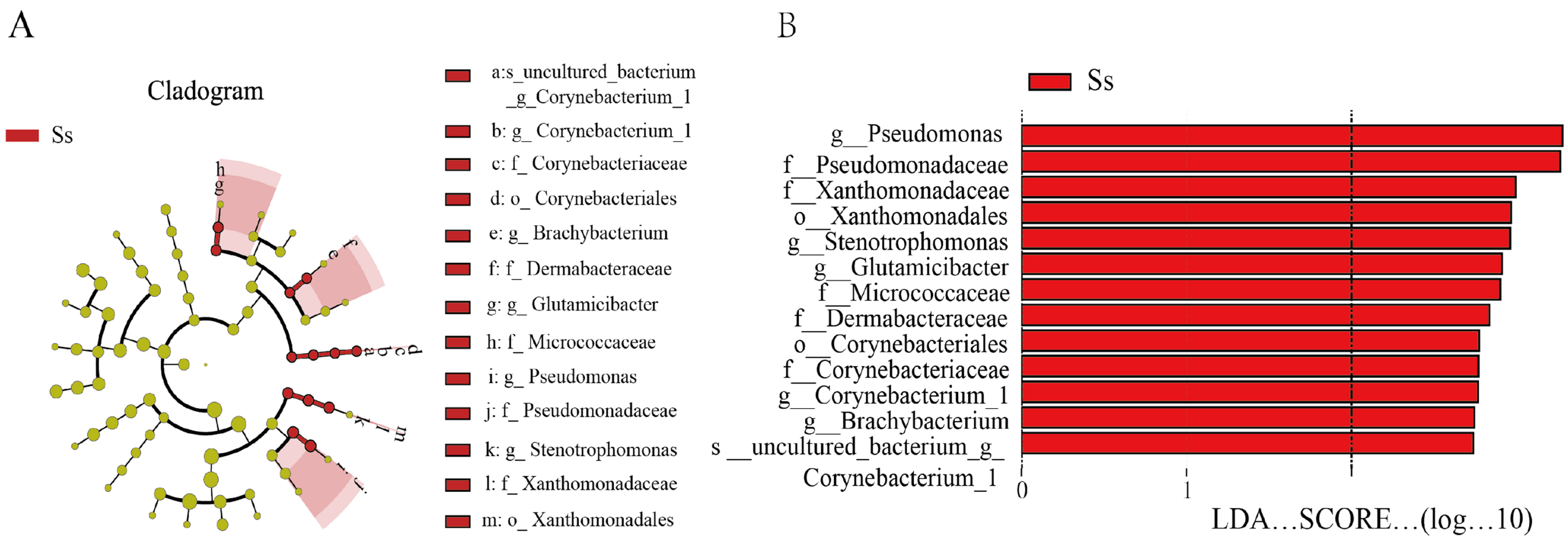

**Figure 5 LEfSe analysis identifies biomarkers that cause differences between groups.** (A) Taxonomic cladogram obtained from LEfSe analysis of 16S sequences and the brightness of each dot is proportional to its effect size. (B) Only taxa meeting an LDA significant threshold >2 are shown.

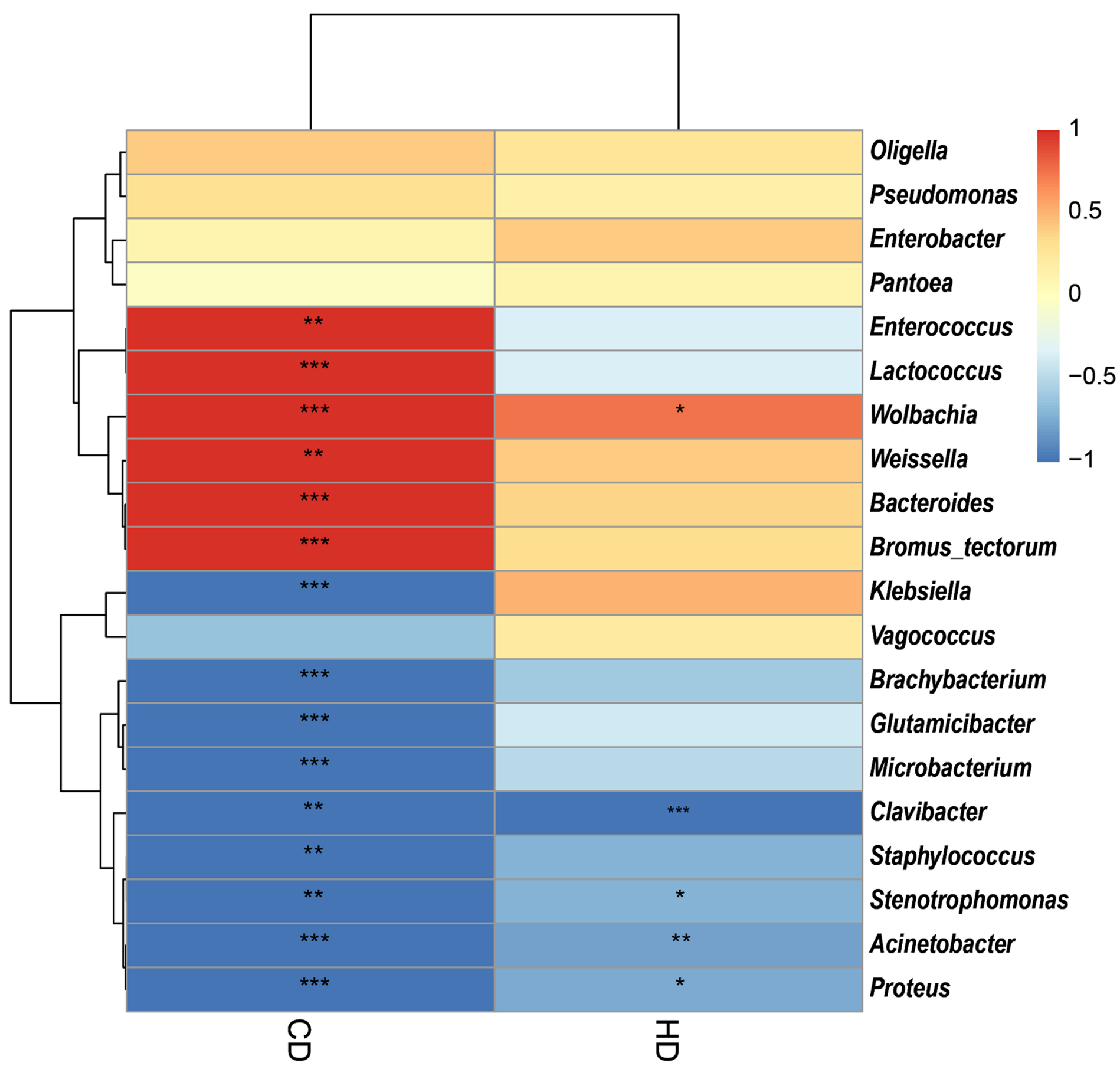

**Figure 6 Heatmap of the correlation between digestibility and bacterial abundance.** *There is a significant correlation of 5% between digestibility and bacteria. **There is a significant correlation of 1% between digestibility and bacteria. ***There is a significant correlation of 0.1% between digestibility and bacteria.

suggesting that these bacteria can be used as candidate bacteria for cellulose and hemicellulose degradation.

## DISCUSSION

In this experiment, we constructed a 16S rRNA gene library via Illumina MiSeq sequencing and applied it to systematically study the intestinal microflora composition of three grasshopper species for the first time. Among the different grasshopper species, the abundance and diversity of intestinal microorganisms were varied. Through the analysis of α and β diversity, it was found that the diversity of the intestinal microflora in the same species was quite varied depending on the collection time. The grasshoppers in this study were collected from summer and autumn populations from the same location, which meant that there were changes in the host insect habitat. Previous studies have shown that the environmental conditions of the habitat of the host insect can affect the interaction between insects and their symbiotic microorganisms, as well as the species and distribution of symbiotic microorganisms (*Schmid et al., 2015*), indicating that the diversity and function of microorganisms in the intestinal tract of insects are closely related to the habitat conditions in which the insects live. However, there are few reports on whether changes in the environment of host insects affects the species and community composition of intestinal microorganisms and the specific extent of that impact, which is a problem worthy of further study.

There were differences in the primary intestinal bacteria among the different species of grasshopper, but the abundance of Proteobacteria was the highest in the intestinal bacteria of all three species of grasshopper, followed by Firmicutes. Bacteria of those two phyla accounted for more than 98% of the total intestinal bacteria of the three grasshopper species, which was consistent with previous reports on the primary species of insect microbiomes. Previous studies have shown that Proteobacteria are the primary bacteria in the intestinal tract of many insects: *Schistocerca gregaria* in Orthoptera (*Dillon et al., 2010*), *Acyrthosiphon pisum* in Hemiptera, and *Ectropis obliqua* (*Engel & Moran, 2013*) and *Spodoptera frugiperda* (*Gichuhi et al., 2020*) in Lepidoptera. Among the Lepidoptera insects studied, the primary bacteria in the intestinal tract of *Lymantria dispar*, *Helicoverpa armigera*, *Bombyx mori* and *Plutella xylostella* larvae are Proteobacteria and Firmicutes (*Zhou et al., 2015*; *Priya et al., 2012*; *Broderick et al., 2004*).

There was variation in the primary genera in the intestinal tract of the grasshopper species. These primary genera and other less abundant genera all play important roles in the life activities of host insects. The content of *Klebsiella* in the intestinal tract of the three species grasshoppers in this study was very high, which was similar to the results of *Liu (2012)* on symbiotic bacteria in the intestinal tract of *Locusta migratoria manilensis* using DGGE. The second most abundant bacteria in our study was *Enterococcus*. This result is consistent with the previous results where bacteria were found to be the most abundant in the microflora of *Schistocerca gregaria* (*Lavy et al., 2019*). In addition, previous studies have shown that *Enterococcus* can not only help degrade lignocellulose but can also produce biogenic amines, which have important physiological functions, such as promoting host growth and enhancing metabolism (*Shu, Lu & Xu, 2011*;

*Shil et al., 2014*). *Enterococcus* may encode 1,4-β-cellobiosidase, endoglucanase and β-glucosidase, which are involved in cellulose degradation, and 1,4-β-xylosidase, which is involved in xylan degradation (*Xia et al., 2013*; *Potrikus & Breznak, 1977*; *Warnecke et al., 2007*); these factors have functions relating to food digestion and absorption. However, some other reported cellulose-degrading bacteria, such as *Enterobacter* and *Pseudomonas* (*Bayer, Shoham & Lamed, 2006*; *Muhammad et al., 2017*), have no significant correlation with cellulose and hemicellulose degradation, and the presence of these bacteria may contribute to eliminating the differences in cellulose digestibility in this study. *Acinetobacter* participates in host food digestion, degrades harmful compounds, and plays a role in nitrogen transformation (*Briones-Roblero et al., 2016*; *Liu et al., 2016*; *Mason et al., 2016*). Existing studies have shown that *Wolbachia* plays an important role in the reproductive ability of host insects (*Hancock et al., 2011*), and whether this genus has an effect on the reproduction of grasshoppers requires further attention.

Since the main food source of grasshoppers is cellulose, it is speculated that the intestinal tract of grasshoppers may contain abundant microorganisms capable of degrading cellulose. *Willis et al. (2010)* isolated cellulase from the foregut and hindgut of the Carolina wasp *Dissosteira carolina*, which was highly similar to the β-1,4-endonuclease of bacteria, fungi and invertebrates, including that secreted by the insects themselves. *Wang et al. (2010)* isolated and screened 5 strains of bacteria with cellulose degradation function from the intestinal tract of *Yunnanacris yunnaneus*, including 4 strains of *Bacillus* and one strain of *Pseudomonas*, which had CMC and filter paper enzyme activities of 167 μ/mL and 9.8 μ/mL, respectively.The above studies show that grasshoppers have the ability to degrade cellulose efficiently. In this study, the contents of cellulose and hemicellulose in the wheat seedling and feces of three species of grasshoppers adults were detected by colorimetry, and the decomposition rates of cellulose and hemicellulose were calculated and analyzed. The cellulose digestibility in *Aiolopus tamulus* and *Oedaleus decorus* was significantly higher than that of hemicellulose. On one hand, this relates result to the structure and composition of cellulose and hemicellulose. Compared with cellulose, hemicellulose has a very complex structure and composition, including xylose, arabinose, mannose and galactose, etc. In the cell wall, hemicellulose is distributed among many celluloses, embedded in the surface of cellulose microfibers and mixed with cellulose. Therefore, only when cellulose is hydrolyzed can hemicellulose be completely hydrolyzed (*Vargas, Weiss & Mcclements, 2007*). On the other hand, the difference in cellulose and hemicellulose digestibility relates to the type and quantity of microorganisms in the grasshopper's intestinal tract. Intestinal microorganisms can secrete a variety of cellulose digestive enzymes. The activities of cellulase and hemicellulase determine the grasshopper's ability to digest cellulose and hemicellulose.

The cellulose digestibility in the three species of grasshoppers was not significantly different, 43.95%, 38.01% and 44.12%, respectively. In a previous study, *Li et al. (2000)* found that the digestibility of crude fiber in different components of corn straw fed to sheep varied from 34.21–61.21%. *Fang, Kuang & Niu (2009)* studied the utilization rate of different straw diets in Xinjiang cattle and found that the digestibility of acid detergent fiber (ADF) and neutral detergent fiber (NDF) in wheat straw was 35.02% and 43.86%,

respectively, and that the digestibility of ADF and NDF in corn straw was 44.26% and 51.91%, respectively (*Fang, Kuang & Niu, 2009*). Meanwhile, a study by *Zhao (2015)* found that the digestibility of cellulose in corn straw by *Locusta migratoria manilensis* was 15.10%. Our results showed that the cellulose digestibility in the three species grasshoppers was significantly higher than that of *Locusta migratoria manilensis* and was close to that of mammals. Whether this difference was related to a difference in the composition of the feeding material needs to be further studied. However, in terms of cellulose digestibility, the intestinal capacity of grasshoppers is very small compared with that of mammals, but their cellulose decomposition rate is close to that of mammals, which indicates that the ability of *Aiolopus tamulus*, *Oedaleus decorus asiaticus* and *Shirakiacris shirakii* to digest cellulose is indeed strong and that studying the cellulose decomposition rate of grasshoppers may be of great value to the development of a cellulose decomposition bioreactor.

Herbivorous insects usually do not directly digest cellulose, or minimally digest cellulose, but mainly digest starch, sugar and protein in food (*Douglas, 2009*). Moreover, the honey bee gut microbiota digests complex carbohydrates, such as hemicellulose and pectin, thereby acquiring energy (*Zheng et al., 2019*). These insects are mainly limited by nitrogen intake rather than carbon source (*McNeil & Southwood, 1978*). *Klebisella* plays an important role in ammonia assimilation into amino acids (*Senior, 1975*), and its negative correlation with the digestion rate may be related to this. Similarly, *San et al. (2011)* also identified some other bacteria related to nitrogen metabolism, including *Staphylococcus*, *Stenotrophomonas*, etc. However, it cannot be ignored that this study is consistent with previous studies, that is, grasshoppers have a strong ability to digest cellulose (*Su et al., 2014*), how much of which is due to the action of their own digestive enzymes and how much of which is due to the contribution of microorganisms needs to be further explored.

Yet, it remains to be seen whether cellulose/hemicellulose digestion in these grasshoppers is exclusively intrinsic or mediated by a combination of intrinsic and bacterial-mediated processes. Until now, there has been no direct evidence that grasshoppers rely entirely on gut microbes to break down cellulose and hemicellulose. For herbivorous insects, the efficiency of decomposition and utilization of cellulose and hemicellulose are largely dependent on gut microbes (*Jehmlich et al., 2016*). *Corynebacterium* and *Glutamicibacter* have been identified from the intestinal bacteria of *Shirakiacris shirakii*. And *Corynebacterium* has been reported to be able to hydrolyze hemicellulose (*Buschke, Schröder & Wittmann, 2011*). *Glutamicibacter* isolated from the intestinal tract of *Proisotoma ananevae* has strong cellulose degradation ability (*Wang et al., 2018*). *Clavibacter* produces cellulase (*Waleron et al., 2010*) and *Brachybacterium* can degrade cellulose (*Zhang et al., 2007*), which supports the results of our correlation analysis. Many insects have intrinsic cellulases (*Davison & Blaxter, 2005*), and some insects belonging to Acrididae have cellulase that can break down plant cell walls (*Calderon-Cortes et al., 2012*). Combined with the results of this article, we can slate a new hypothesis: the intestinal microorganisms of grasshoppers have a great influence on the decomposition of cellulose/hemicellulose.

## CONCLUSIONS

This study analyzed the intestinal microbial diversity of three species of grasshoppers, using the method of 16S rDNA gene library construction. Proteobacteria and Firmicutes are the dominant bacteria in the intestinal microbial communities of the three grasshoppers species. However, the dominant genera of different species grasshoppers are different. *Shirakiacris shirakii* had the highest bacterial species richness, and *Aiolopus tamulus* had the highest bacterial diversity. The intestinal microflora structure varied between the different species of grasshoppers, with the intestinal microflora structure of *Aiolopus tamulus* and *Shirakiacris shirakii* being the most similar. Meanwhile, the time at which grasshopper specimens were collected also led to changes in the intestinal microflora structure in the same species of grasshoppers.

There was no significant difference in cellulose digestibility between the three species of grasshoppers ($P > 0.05$), while the hemicellulose digestibility of *Shirakiacris shirakii* was significantly higher than *Aiolopus tamulus* and *Oedaleus decorus asiaticus* ($P < 0.05$). In addition, the cellulose digestibility of *Aiolopus tamulus* and *Oedaleus decorus asiaticus* was significantly higher than the hemicellulose digestibility.

LEfSe analysis and Spearman correlation coefficients showed that the hemicellulosic digestibility of *Shirakiacris shirakii* was significantly higher than that of the other two species of grasshopper, which may be related to the presence of *Pseudomonas, Stenotrophomonas*, *Glutamicibacter*, *Corynebacterium* and *Brachybacterium* in *Shirakiacris shirakii* intestinal tract.

This study lays a foundation for the utilization of garsshoppers intestinal microorganisms in the future.

## ACKNOWLEDGEMENTS

We would like to thank DBMediting for professional English language editing services.

### Funding

This study is funded by the Natural Science Foundation of Hebei Province (No. C2018201139), the Natural Science Foundation of China (Nos. 31872274, 31702043 and 32070473), and the Post-graduate's Innovation Fund Project of Hebei University (No. hbu2019ss022). There was no additional external funding received for this study. The funders had no role in study design, data collection and analysis, decision to publish, or preparation of the manuscript.

### Grant Disclosures

The following grant information was disclosed by the authors:
Natural Science Foundation of Hebei Province: C2018201139.
Natural Science Foundation of China: 31872274, 31702043 and 32070473.
Post-graduate's Innovation Fund Project of Hebei University: hbu2019ss022.

## Competing Interests

The authors declare that they have no competing interests.

## Author Contributions

- Jian-Mei Wang conceived and designed the experiments, performed the experiments, analyzed the data, prepared figures and/or tables, authored or reviewed drafts of the paper, and approved the final draft.
- Jing Bai conceived and designed the experiments, performed the experiments, prepared figures and/or tables, and approved the final draft.
- Fang-Yuan Zheng analyzed the data, prepared figures and/or tables, authored or reviewed drafts of the paper, and approved the final draft.
- Yao Ling analyzed the data, prepared figures and/or tables, and approved the final draft.
- Xiang Li performed the experiments, prepared figures and/or tables, and approved the final draft.
- Jing Wang performed the experiments, prepared figures and/or tables, and approved the final draft.
- Yong-Chao Zhi conceived and designed the experiments, analyzed the data, authored or reviewed drafts of the paper, and approved the final draft.
- Xin-Jiang Li conceived and designed the experiments, authored or reviewed drafts of the paper, and approved the final draft.

## Data Availability

The raw data are available in the Supplemental Files and the gene sequencing data are available at GenBank: PRJNA635459.

## Supplemental Information

Supplemental information for this article can be found online at http://dx.doi.org/10.7717/peerj.10194#supplemental-information.

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
