# Peer review of "Diversity of the gut microbiome in three grasshopper species using 16S rRNA and determination of cellulose digestibility"

_PeerJ, doi:10.7717/peerj.10194_

## Round 0.1 · original submission · Major Revisions

Although this paper presents some new information on the gut bacterial community of grasshoppers, I agree with the reviewers that several aspects need to be improved before the manuscript can be accepted. The authors should think about the review comments and revise the manuscript carefully. Especially, you should pay much attention to: 1) the clarity of your methods and experiments (as one reviewer suggested, you should make several places of the methods clearer); 2) the reliability of your claim and discussion (any discussion should be based on direct evidence); 3) the presentation and language of the manuscript. Too many figures and tables are not good, instead, you need to assemble related figures into multiple-panel figures following a logical way, and some figures with less information can be provided as supplementary files. Actually separate tables can also be incorporated. And, the language needs to be improved.

I suggest the authors revise the manuscript carefully. The decision of Major Revision can not guarantee future acceptance if you cannot handle the reviewers' comments in a proper way. You should also provide a file with point-to-point responses to the review comments when submitting revision files.

Reviewer 1 ·

Basic reporting

Grasshoppers are the notorious pests for some agricultural crops. It is of great significance to determine the potential role of gut bacteria in the utilization of plant hosts for insect pests. Although the authors have presented some interesting findings here, this manuscript can not be accepted for publication now because of some visible defects. Firstly, the English language should be improved significantly because there are many grammar mistakes and awkward sentences. Secondly, the results of this manuscript were not well summarized and presented. The data were described too generally, but the major results were not summarized concisely and directly. Therefore, it is very difficult to catch up what the major findings the authors presented here. Thirdly, some important progress has been done on the interactions between insect pests and their gut microbiota. However, the authors did not pay sufficient attention to these achievements on this topic in 2018-2020. Moreover, many sentences on describing experimental methods can be found in the section of results. It is better to remove them because this section should often focus on what you have found by your experiments. Finally, the authors did not present any direct evidence to support the cellulose digestibility of three grasshoppers was from their gut bacteria. But the authors did only do some correlation analysis and try to support their view. In my opinion, it is not enough to support this finding. It is better if the authors can isolate and identify some cultivable bacterial species from the guts of three grasshopper species.

Experimental design

Some important experimental information missed in this manuscript. For example, the authors said they collected the adults of three grasshopper species in field and fed with wheat seedlings in lab. However, the authors did not introduce which wheat species or variety was used to feed these insects. Furthermore, the authors did not tell how long these collected insects were fed in lab before the following experiments and the conditions that were used to rear these insects in lab. Last, the authors did not tell the statistical method and how they complete the statistical analysis.

Validity of the findings

The major findings of this manuscript were not summarized and presented well. Because of the absence of some novel important publications, the findings of this manuscript were not well explained in the section of discussion. The authors should clearly present what great differences on gut microbiota among these three grasshopper species and what impact of these differences in their gut bacteria on the cellulose digestibility.

Additional comments

Some format questions on reference can be easily found in the text and reference list of this manuscript. Therefore, the authors should correct these questions carefully.

Reviewer 2 ·

Basic reporting

no comment

Experimental design

no comment

Validity of the findings

no comment

Additional comments

This paper describes the gut microbiome of three grasshopper species, and attempts to correlate the differences in microbiota structure with the ability of the host to digest cellulose and hemicellulose. The data is potentially interesting and should be useful for subsequent studies of grasshopper gut bacteria. However, it should be noted that this paper represents only very weak association between change in gut bacteria and the food digestion. Without further data showing the function of gut microbiota in helping digesting cellulose, it is difficult to justify the rational of the current study. Therefore, the authors need to do a better job with any claims that the difference in several bacterial species is actually causing the difference in host’s digestion. For example, the claim that the hemicellulose digestibility of Shirakiacris shirakii was significantly higher than that of the other two species of grasshopper DUE TO the presence of Psudomonas, Stenotrophomonas, Glutamicibacter, Corynebacterium, and Brachybacterium in its intestinal tract, was not supported by the current analysis, and needs to be re-phrased.
Numerous language problems have to be fixed first to make the paper readable, probably by sending to a native English speaker to check. Finally, It is strongly suggested that the authors assembly the figures into multiple-panel figures.

---

## Round 0.2 · Minor Revisions

The authors have significantly improved the manuscript and addressed most of the concerns from reviewers. For example, they added rearing conditions and tests used in statistical analysis to make the Methods clearer, and they improved the presentation by adopting multiple-panel figures. However, some minor problems still remain to be addressed.

1. The title of the manuscript can be changed to "Diversity of the gut microbiome in three grasshopper species using 16S rRNA and determination of cellulose digestibility".

2. The authors mentioned in the Results that Spearman correlation was calculated (Line 320-321), they should state how they did the correlation analysis in the Methods part.

3. Currently, the content of the Introduction is suitable, but the authors should improve the diversity and representativeness of citations in the Introduction by citing more references from high-profile journals and international authors.

4. There are still some minor format problems in the manuscript, for example in the reference list.

Reviewer 1 ·

Basic reporting

Grasshoppers are the notorious pests for some agricultural crops.The authors presented some interesting findings on the gut microbiota of three grasshopper species and gut bacteria confer the cellulose digestibility to help the host to achieve nutrients. This manuscript has been improved significantly and it can be accepted for publication now.

Experimental design

no comment

Validity of the findings

no comment

Additional comments

This manuscript has been improved significantly and it can be accepted for publication now. However, there are still some small problems on the journal names in the reference list. Please corrected them before publication.

---

## Round 0.3 · accepted · Accept

The issues raised by reviewers and I have been addressed. I think the manuscript can be accepted for publication now.